# Trends in Obesity and Obesity-Related Risk Factors among Adolescents in Korea from 2009 to 2019

**DOI:** 10.3390/ijerph19095672

**Published:** 2022-05-06

**Authors:** Eunok Park, Young Ko

**Affiliations:** 1College of Nursing·Health and Nursing Research Institute, Jeju National University, Jeju 63243, Korea; eopark@jejunu.ac.kr; 2College of Nursing, Gachon University, Incheon 21936, Korea

**Keywords:** adolescent, obesity, physical activity, diet, trends

## Abstract

This study was conducted to identify the trends in obesity prevalence among adolescents and changes in the risk factors related to obesity. The study analyzed secondary data obtained from Korean Youth Risk Behavior Web-based Surveys conducted from 2009 to 2019. The Korean Youth Risk Behavior Web-based Survey is an annual survey of a nationwide representative sample of enrolled students aged 13–18 years in middle and high schools in Korea. Linear and trend analyses showed that the prevalence of obesity increased by 0.47% on average annually; this increase was statistically significant. Healthy food intake decreased significantly, but the prevalence of unhealthy food intake and the prevalence of skipping breakfast increased significantly. Vigorous-intensity physical activity, physical activity for over 60 min a day, and muscle-strengthening exercise for more than 3 days a week increased significantly, but so did the adolescents’ sedentary time. Therefore, health care providers and public policymakers need to actively manage adolescent obesity, which has been continuously increasing since 2009. In addition, long-term trends in obesity-related risk factors such as physical activity and dietary behaviors need to be considered in the development of obesity management strategies.

## 1. Introduction

Adolescent obesity is associated with several health problems such as high blood pressure, high cholesterol, breathing problems, impaired glucose tolerance, joint problems, and fatty liver disease [1]. Additionally, adolescents with obesity can suffer psychological issues such as depression; anxiety; and poor self-esteem, body image, and peer relationships [2,3]. Obesity threatens the healthy growth of adolescents and lowers their quality of life [4]. Moreover, adolescent obesity is associated with the risk of adult obesity, cardiovascular morbidity, and mortality [5]. Thus, adolescent obesity can be considered an important public health issue.

Several studies on obesity trends in the 2000s have shown increasing rates of obesity among adolescents [6,7]. In particular, the prevalence of obesity among Korean adolescents increased from 10.7% in 2006 to 12.2% in 2012–2015; the increase was higher for boys than girls [7]. However, recent studies have supported the decreasing or stabilizing trends of adolescent obesity prevalence in some developed countries [8] and European countries [9,10]. Since the trends in obesity prevalence can change and differ from country to country and time to time, we need more information on the time trends of obesity among Korean adolescents to establish public health policies and intervention strategies for timely obesity management.

Many previous studies have reported the relationship between obesity and its risk factors [11,12,13,14] and trends in the prevalence of obesity [6,7,8,9,10,11]; however, few studies have been conducted to identify trends in obesity-related risk factors, such as physical activity and dietary behavior, along with trends in obesity prevalence. Therefore, our study aimed to identify the trends in obesity prevalence and the trends in physical activity and dietary behaviors among Korean adolescents, using nationally representative data from 2009 to 2019. The results of this study can provide important basic data for planning future projects for adolescent obesity prevention and management interventions.

## 2. Materials and Methods

### 2.1. Study Design and Participants

This study basically consisted of a secondary data analysis using data from Korean Youth Risk Behavior Web-based Surveys (KYRBSs) conducted from 2009 to 2019 every year. The KYRBS was introduced in 2005 jointly by the Korean Ministry of Education, Korean Ministry of Health and Welfare, and Korea Centers for Disease Control and Prevention (KCDC). The KYRBS is an annual survey of a nationwide representative sample of enrolled students aged 13–18 years in middle and high schools in Korea [15,16]. The sampling method was stratified random cluster sampling. The target population was all public and private middle and high school students, and the sampling has performed every year. First, schools were chosen as primary sampling units using a random sampling method based on population stratification and sample allocation; then, one class of each grade at the sampled schools were selected using a random sampling method each year [15,16]. In this study, we used the publicly available data of the 4th–14th KYRBSs published from 2009 to 2019 (Table 1 shows the number of participants for each year) [15]. We used all data for this study.

### 2.2. Measurements

#### 2.2.1. Obesity

In each survey, the most recently measured weight and height are self-reported, and the BMI is calculated using these data. We used the percentile-based cutoff points for obesity which were presented as recommended criteria by the Korean Pediatric Society [17]. Obesity is then defined at or above the sex-specific 95th percentile on the BMI-for-age growth charts developed by the Ministry of Health and Welfare and the Korean Academy of Pediatrics [17]. The Korean growth chart is a percentile curve showing the distribution of body measurements such as the height and weight of children and adolescents in Korea. It is used as an index to evaluate the growth status of children and adolescents in Korea, such as short stature, underweight, and obesity [18]. This chart has been published every 10 years since 1967. For our study, we used the Child and Adolescent Growth Chart published in 2017 [17,18].

#### 2.2.2. Dietary Behaviors and Physical Activity

Dietary behaviors in the survey included skipping breakfast and the intake of fruits, vegetables, milk, fast food, carbonated drinks, sweet drinks, and energy drinks.

Skipping breakfast was defined as not having breakfast for more than 5 days in a week. This is assessed by asking the question, “Over the past 7 days, on how many days did you have breakfast?” Fruit intake was assessed as having fruit at least once a day (excluding fruit juice) in a week. Vegetable intake was assessed by asking the question, “Over the past 7 days, how many times a day did you have vegetables (excluding potatoes)?” The respondents were then divided into two groups: those who had vegetables at least three times a day and those who had vegetables less than three times a day. Milk intake was defined as drinking milk (both fresh and processed milk, with or without sugar) at least once a day in a week. This is assessed by asking the question, “During the past 7 days, how often did you drink milk (including white and processed milk)?”

Fast food intake was assessed as having fast food at least three times a week. The intake of carbonated drinks was measured as having carbonated drinks at least three times a week. Sweet drink intake was assessed by asking the question, “Over the past 7 days, how often did you have sweet drinks?” The respondents were then divided into two groups: those who had sweet drinks at least three times a week and those who had sweet drinks less than three times a week. Energy drink intake was assessed as having energy (or high caffeine) drinks at least three times a week.

Physical activity included vigorous-intensity physical activity, 60 min of physical activity per day, muscle-strengthening physical activity, and sedentary time.

Vigorous-intensity physical activity was assessed by asking the question, “Over the past 7 days, on how many days did you carry out high-intensity physical activities (such as jogging, soccer, basketball, taekwondo, mountaineering, fast cycling, fast swimming, and carrying heavy objects) for 20 min or more?” The respondents were then divided into two groups, those who did vigorous-intensity physical activity on 3 days or more and those who done so on fewer than 3 days [19,20].

Sixty minutes of physical activity per day was assessed by asking the question, “Over the past 7 days, on how many days did you do physical activities that increased your heart rate from normal for 60 min or more, combining all your physical activity?” The respondents were divided into two groups: those who did so on 5 days or more and those who did so on fewer than 5 days. Muscle-strengthening exercises were assessed by asking the question, “Over the past 7 days, on how many days did you carry out muscle-strengthening exercises such as push-ups, sit-ups, weightlifting, dumbbells, iron bars, and parallel bars to build muscle strength for at least 10 min?” The respondents were categorized into two groups; those who did so on 3 days or more and those who did so on fewer than 3 days.

Sedentary time was measured by asking the question, “Over the past 7 days, how many hours per day on average did you spend sitting?” The respondents were asked to divide their sedentary time into four sub-items: time spent sitting for study purposes on weekdays, sitting for purposes other than study on weekdays, sitting for study purposes on the weekend, and sitting for purposes other than study on the weekend. The sum of all four sub-items was considered their sedentary time.

The validity and reliability of the survey items of dietary behaviors and physical activity of the KYRBS questionnaire were evaluated in a previous study [21,22,23]. The questions on dietary behaviors and physical activity showed moderate reliability and validity [22], and the Food Frequency Questionnaire in Adolescents (FFQA) also reported moderate reliability and validity [21,22].

### 2.3. Data Collection

The KYRBS protocol was approved by the KCDC and Prevention Institutional Review Board (IRB) annually up to 2014. IRB approval was not required from 2015 onward in view of the Population Health Promotion Act 19 (approval number 117058). In 2016 and 2017, the Research Ethics Review Committee (RERC) of the KCDC decided that the KYRBSs corresponded to a study conducted by the state for public welfare. Therefore, in view of the opinion that one could investigate without RERC approval, we collected data that were not reviewed by the RERC [16]. The KYRBS protocol was again approved by the KCDC IRB in 2019. The participants take part in this web-based self-reported survey voluntarily under the principles of anonymity in a school computer laboratory [16]. Written informed consent (both directly and through their parents or legal guardians) was obtained from all participants before participation. In compliance with the Privacy Act and the Statistics Act, we could only obtain non-identifying data, so that participants could remain anonymous [16]. We used raw data released for academic purposes from the website https://knhanes.kdca.go.kr (accessed on 1 October 2020. The study protocol was approved by the IRB of the university (IRB No. 1044396-202109-HR-197-01) with which the researchers were affiliated. Informed consent was waived by IRB.

### 2.4. Statistical Analysis

We integrated the data according to raw data analysis guidelines, designing and analyzing complex sample design elements using the SAS program. All prevalence rates were obtained applying sampling weight for each year. Missing data were excluded when analyzing prevalence rate, and the number of missing data were varied over the variables and the years. We conducted the trend analysis method to estimate the annual prevalence change using the prevalence rate. For estimating the annual prevalence change and testing the statistical significance of the annual change, we carried out simple linear regression analysis setting ‘year’ as an independent variable and ‘prevalence rate’ as a dependent variable. In the case of not having prevalence in certain years due to changes to the survey questionnaire, the estimated annual change was calculated with the available prevalence rate.

## 3. Results

### 3.1. Trends in Obesity Prevaelnce

The trends in obesity prevalence among Korean adolescents are shown in Table 2. The prevalence of obesity among Korean adolescents doubled from 5.1% in 2009 to 11.1% in 2019; the increase was from 6.6% in 2009 to 13.8% in 2019 for boys and from 3.5% in 2009 to 8.1% in 2019 for girls. The prevalence of obesity showed a statistically significant increase of 0.47% for all years (total, 0.47%; boys, 0.56%; girls, 0.38%).

### 3.2. Changes in Dietary Behaviors and Physical Activity

The changes in dietary behaviors of adolescents from 2009 to 2019 are shown in Table 3. Overall, the prevalence of skipping breakfast for more than 5 days a week increased by 0.51% annually, rising from 27.1% in 2009 to 35.9% in 2019. For boys, skipping breakfast increased by 0.45% annually, rising from 28.2% in 2009 to 34.6% in 2019. For girls, skipping breakfast increased by 0.57% annually, rising from 25.9% in 2009 to 36.9% in 2019.

The healthy food intake of adolescents has been showing a significant decrease. The intake of fruits more than once a day decreased, falling from 24.7% in 2009 to 20.5% in 2019, resulting in a 0.88% decrease in fruit intake annually. Vegetable intake more than three times a day also decreased, from 17.9% in 2009 to 10.9% in 2019. The habit of drinking milk more than once a day showed a significant decrease of 1.12% annually. In contrast, unhealthy food intake increased significantly: fast food intake more than 3 days a week increased from 12.1% in 2009 to 25.5% in 2019; carbonated drink intake more than three times a week increased from 24.0% in 2009 to 37.0% in 2019; sweet drink intake more than 3 times a week increased from 38.2% in 2014 to 50.4% in 2019; and energy drink intake more than 3 times a week increased from 3.3% in 2014 to 12.2% in 2019.

Boys showed a significant decrease in healthy food intake. Their fruit intake decreased from 24.0% in 2009 to 20.3% in 2019, with a 0.82% decrease in fruit intake annually, and vegetable intake decreased from 18.9% in 2009 to 12.4% in 2019. The milk intake of boys decreased by a significant 1.35% annually. Furthermore, the unhealthy food intake of boys increased significantly; their fast food intake increased from 13.4% in 2009 to 27.5% in 2019, carbonated drink intake increased from 29.8% in 2009 to 45.1% in 2019, sweet drink intake increased from 41.7% in 2014 to 53.6% in 2019, and energy drink intake increased from 4.3% in 2014 to 12.8% in 2019.

Girls also showed a significant decrease in healthy food intake. Their fruit intake decreased from 25.5% in 2009 to 20.6% in 2019 (b = −0.95, *p* < 0.001) and vegetable intake decreased from 16.9% in 2009 to 9.3% in 2019. The milk intake of girls also decreased significantly (b = −0.85, *p* < 0.001). Furthermore, their unhealthy food intake increased significantly: their fast food intake increased from 10.7% in 2009 to 23.4% in 2019, carbonated drink intake increased from 17.4% in 2009 to 28.1% in 2019, sweet drinks intake increased from 34.4% in 2014 to 47.0% in 2019, and energy drink intake increased from 2.2% in 2014 to 11.6% in 2019.

The changes in adolescents’ physical activity from 2009 to 2019 are shown in Table 4. All adolescents, both boys and girls, showed a significant increase in vigorous-intensity physical activity, physical activity for over 60 min a day, muscle-strengthening exercises for more than 3 days a week, and sedentary time in trend analysis. For all adolescents, the prevalence of vigorous-intensity physical activity over 20 min a day for 3 days a week increased from 10.9% in 2009 to 14.7% in 2019; 32.0% reported that they engaged in physical activity for 60 or more minutes a day 3 days a week in 2019. Similarly, 21.9% reported that they did muscle-strengthening exercises for 10 or more minutes 3 or more days a week in 2019. The adolescents’ sedentary time also increased, from 1057.2 min in 2013 to 1160.2 min in 2019.

For boys, the table reports that their vigorous-intensity physical activity for 20 min a day 3 or more days a week increased from 15.7% in 2009 to 21.5% in 2019. Furthermore, 44.8% of boys reported that they engaged in physical activity for 60 or more minutes a day 3 days a week in 2019. Similarly, 33.4% reported that they carried out muscle-strengthening exercises for 10 or more minutes a day 3 or more days a week in 2019. Their sedentary time also increased, from 973.8 min in 2013 to 1099.0 min in 2019. For girls, the prevalence of vigorous-intensity physical activity was 5.4% in 2009 and 7.3% in 2019. The percentage of girls reporting physical activity for 60 or more minutes 5 days a week was 18.4% in 2009 and 18.0% 2019, with the trend showing a 0.50% increase annually. The increase in muscle-strengthening exercises was not significant for girls. However, their sedentary time increased from 1148.5 min in 2013 to 1225.8 min in 2019.

## 4. Discussion

This study identified the trends in adolescent obesity prevalence, dietary behaviors, and physical activity from 2009 to 2019 using nationally representative sample data. First, the prevalence of obesity among adolescents doubled from 5.1% in 2009 to 11.1% in 2019, increasing significantly by 0.47% on average annually. According to previous studies, the average obesity prevalence among Korean children and adolescents has been increasing since 1997, showing a remarkable increase for boys [7,24]. The findings of this study indicate that the prevalence of obesity among adolescents in Korea differs from that in some other developed countries, where studies have shown stable or decreasing obesity rates [11]. Studies analyzing the trends of obesity among Chinese adolescents between 1991 and 2015 have shown a decline in obesity rates since 2011, unlike in Korea [25]. This may indicate that the prevention and management of obesity among adolescents in Korea has not been successful, even though several interventions have been implemented through various strategies at the national and school levels. The obesity rate of Korean adolescents is increasing compared to several other countries, but it is not high compared to some European countries [26] or the United States [27]. However, as the obesity of adolescents continues to increase, we need to pay attention to this issue among Korean adolescents and address the causes of the increase in obesity.

Second, the prevalence of obesity showed a significant increase in both boys and girls. Although we did not compare the prevalence of obesity between boys and girls statistically, the prevalence of obesity was higher in boys than in girls for all years, and the year-over-year increase was greater in boys than in girls. According to the World Health Organization, the prevalence of overweight and obesity among children and adolescents aged 5 to 19 years has increased sharply from 4% in 1975 to 18% in 2016; this increase is similar for boys and girls (18% for girls and 19% for boys) [26]. However, studies carried out in some countries have reported differences in trends in the prevalence of obesity among boys and girls [6,28]. The lower prevalence of obesity in girls than in boys in this study may be due to self-reported measures of height and weight. Several researchers reported that boys and girls tend to report their own weight less than their measured weight and girls tend to report their weight much less than boys [29,30,31]. In addition, this study was to identify the 10-year trend analysis using measured data from a new sample every year, and it was not possible to identify various factors affecting the trends in the prevalence of obesity in boys and girls. Therefore, it is necessary to conduct future research to confirm the trend difference in the prevalence of obesity between boys and girls using data from direct measures and to identify factors contributing to this trend difference.

Third, all types of diet behaviors among both boys and girls continued to change for the worse from 2009 to 2019. These changes in dietary behaviors may be associated with the increase in obesity rates among adolescents. Some studies have reported an increased association between western or modern diet trends and obesity in Korea [12], China [32], and Asian developing countries [33]. In particular, girls show a higher tendency of skipping breakfast than boys. Breakfast skipping has been reported to be associated with adolescent obesity in studies conducted in India and the United States [14,34]. In fact, breakfast skipping is related to not only the food consumed for breakfast but also health value [14]. The increased rate of skipping breakfast among girls may indicate fewer health concerns. On the other hand, boys tend to show a higher inclination toward unhealthy behaviors than girls, especially regarding fast food and carbonated drink intake. A systematic review of the relationship between adolescent obesity and diet has shown that a diet with a lower percentage of obesogenic foods would be effective in preventing obesity [35]. Therefore, we need to intervene the unhealthy eating behaviors of adolescents and educate them on healthy dietary behaviors. Due to the limitations of the data obtained by collecting data from a new sample every year, this study could not identify which dietary behavior changes were related to an increase in the prevalence of obesity. Therefore, the relationship between specific dietary behavior and obesity needs to be reconfirmed.

We found the positive result that the rate of physical activity is increasing among Korean adolescents. These results of this study differ from the findings of the 2001–2016 international trend study showing increasing insufficient physical activity among boys [36]. The physical activity rate of Korean boys was higher than the average rate of 15-year-olds in the European Union (EU), whilst the rate of physical activity among Korean girls was at the EU level [24]. In an 8-year trend analysis of American adolescents, an increase in exercise for more than 60 min a day was associated with a decrease in obesity [33]. Therefore, the continuous increase in physical activity among Korean adolescents is expected to have a positive effect on obesity prevention and management.

However, this study shows a significant increase in the sedentary time of participants; this is consistent with the results of previous studies [33,37,38]. Although the sedentary time of adolescents has been gradually increasing, the relationship between sedentary lifestyles and obesity has not yet been proven [39]. As the increase in sedentary lifestyles of Korean adolescents may contribute to increased obesity rates in the future, we need to continuously monitor the relationship between increased sedentary time and obesity among Korean adolescents.

This study’s findings are significant in that the trends of dietary behaviors and physical activity as well as the prevalence of obesity have been confirmed using nationally representative data. However, this study has some limitations. First, our measurements of weight and height have been taken from self-reported data. Although the self-reported measures of weight and height have been validated in epidemiological studies to identify overweight in children and adolescents [40,41], the self-reported measure applied in this study is subject to recall, responses, and social desirability bias, which may have affected the results. In a study of Korean adolescents, it was reported that self-reported height and weight underestimated BMI because they tend to underestimate their weight and overestimate their height, which in turn could lead to a lower prevalence of obesity [42]. These biases should be considered when interpreting the findings of this study using self-reported data to identify prevalence and trends in adolescent obesity. Physical activity and dietary behavior questionnaires used in this study were validated [21,22]. However, questionnaires on dietary behavior and physical activity also have well-known disadvantages (e.g., social desirability, under- or over-reporting). Previous studies found that adolescents’ self-reports on health behaviors such as diet and physical activity were affected by cognitive and situational factors, but not to the extent of threatening the validity [43]. However, some previous studies have reported a difference between self-reported and directly measured values [44]. It is therefore possible that the results of this study obtained through self-reporting may not reflect reality.

Second, this study was designed to identify the long-term trend of obesity, physical activity, and dietary habits using data measured from a new national representative sample every year. Considered caution is needed in interpreting the results, as the causal relationship between changes in physical activity and dietary behaviors and the prevalence of obesity cannot be confirmed in this study. Finally, for the definition of obesity, Korean adolescent obesity criteria (for obesity treatment and management in Korean clinical and school practice) was used. Therefore, there is a limit to comparing the prevalence of obesity, not the tendency of obesity, among adolescents from other countries.

## 5. Conclusions

The prevalence of obesity has increased significantly among Korean boys and girls aged 13–18 years. In addition, this study found an increase in the prevalence of unhealthy diet intake and sedentary time and a decrease in the prevalence of healthy diet intake. These findings imply that health care providers and public policymakers need to actively manage adolescent obesity, which has been continuously increasing since 2009. Trends in obesity-related risk factors, such as physical activity and dietary behavior, need to be considered when developing obesity management strategies. Boys need to reduce their sedentary time and change their unhealthy diets, including fast food and carbonated drink intake, while girls need to increase their physical activity and reduce their intake of sweet and carbonated drinks.

## Figures and Tables

**Table 1 ijerph-19-05672-t001:** The Number of Participants by Sex from 2009 to 2019.

Year	Boys		Girls		Total	
	*n*	%	*n*	%	*n*	%
2009	38,152	52.7	34,247	47.3	72,399	100.0
2010	37,090	52.4	33,719	47.6	70,809	100.0
2011	36,755	50.0	36,719	50.0	73,474	100.0
2012	37,229	51.5	35,000	48.5	72,229	100.0
2013	35,575	50.6	34,779	49.4	70,354	100.0
2014	35,391	50.6	34,568	49.4	69,959	100.0
2015	34,152	51.7	31,916	48.3	66,068	100.0
2016	32,904	51.6	30,837	48.4	63,741	100.0
2017	30,662	50.8	29,730	49.2	60,392	100.0
2018	29,613	50.8	28,723	49.2	58,336	100.0
2019	29,059	52.1	26,689	47.9	55,748	100.0

**Table 2 ijerph-19-05672-t002:** Trends of Obesity Prevalence among Korean Adolescents by Sex.

Variables	2009	2010	2011	2012	2013	2014	2015	2016	2017	2018	2019	b	t	*p*
Overall	5.1	5.3	5.6	6.2	6.6	6.9	7.5	9.1	10.0	10.8	11.1	0.47	7.59	<0.001
Boy	6.6	7.0	6.8	7.5	7.9	8.5	8.8	11.1	12.3	13.4	13.8	0.56	6.83	<0.001
Girl	3.5	3.5	4.2	4.8	5.2	5.2	6.1	6.9	7.6	8.0	8.1	0.38	9.00	<0.001

Note: The prevalence was calculated by applying sampling weight. Annual prevalence from 2009 to 2019 was used for trend analysis. *p*-value refers to the statistical significance of test results for the annual change in prevalence.

**Table 3 ijerph-19-05672-t003:** Trends of Prevalence of Dietary Behaviors of Adolescents from 2009 to 2019 by Sex.

Variables	Category	Year											b	t	*p*
		2009	2010	2011	2012	2013	2014	2015	2016	2017	2018	2019			
Skipping breakfast (more than 5 days/week)	Overall	27.1	25.6	24.4	24.8	26.4	28.5	27.9	28.2	31.5	33.6	35.7	0.51	3.67	0.003
	Boys	28.2	25.5	25.3	24.9	26.7	28.2	26.9	27.3	30.1	32.2	34.6	0.45	3.84	0.002
	Girls	25.9	25.6	23.4	24.6	26.2	28.9	28.9	29.3	33.0	35.1	36.9	0.57	3.25	0.006
Fruit intake (more than 1 time/day)	Overall	24.7	22.9	20.3	18.7	19.7	22.0	22.9	23.2	22.2	20.8	20.5	−0.88	−4.14	0.001
	Boys	24.0	22.2	19.2	17.4	18.8	20.8	22.5	22.9	21.6	20.8	20.3	−0.82	−3.75	0.002
	Girls	25.5	23.7	21.5	20.1	20.8	23.4	23.3	23.5	22.9	20.9	20.6	−0.95	−4.56	0.001
Vegetable ^†^ intake (more than 3 times/day)	Overall	17.9	17.9	17.9	17.1	16.6	15.6	15.3	14.3	14.4	-	10.9	−0.37	−3.92	0.002
	Boys	18.9	18.8	18.7	17.9	17.7	16.6	16.6	15.4	15.9	-	12.4	−0.37	−4.23	0.001
	Girls	16.9	16.8	17.0	16.2	15.4	14.5	13.9	13.2	12.7	-	9.3	−0.37	−3.61	0.004
Drinking ^†^ milk (more than 1 time/day)	Overall	28.7	28.2	28.7	28.9	29.3	26.6	27.8	26.8	25.0	-	22.8	−1.12	−6.74	<0.001
	Boys	34.6	34.3	34.8	35.4	35.7	32.8	34.4	33.6	30.2	-	28.4	−1.35	−6.46	<0.001
	Girls	22.0	21.5	21.9	24.0	22.3	19.9	20.5	19.4	19.3	-	16.7	−0.85	−6.49	<0.001
Fast food intake (more than 3 days/week)	Overall	12.1	12.0	11.6	11.5	13.1	15.6	14.8	16.7	20.5	21.4	25.5	1.30	7.04	<0.001
	Boys	13.4	13.4	13.0	12.6	14.4	16.5	16.0	17.9	21.6	22.7	27.5	1.32	6.54	<0.001
	Girls	10.7	10.4	10.0	10.3	11.6	14.5	13.5	15.4	19.3	20.1	23.4	1.29	7.57	<0.001
Carbonated drinks intake (more than 3 times/week)	Overall	24.0	24.3	23.2	24.3	25.5	26.0	28.3	27.1	33.7	34.7	37.0	1.33	6.56	<0.001
	Boys	29.8	29.8	28.8	30.1	31.6	32.3	35.3	32.5	40.2	41.9	45.1	1.52	6.31	<0.001
	Girls	17.4	18.2	17.0	18.0	18.7	19.1	20.8	21.2	26.7	26.8	28.1	1.14	6.78	<0.001
Sweet drinks intake (more than 3 times/week)	Overall	-	-	-	-	-	38.2	41.9	41.4	47.1	47.1	50.4	2.63	7.04	0.002
	Boys	-	-	-	-	-	41.7	45.5	42.8	49.8	53.6	53.6	2.59	4.74	0.009
	Girls	-	-	-	-	-	34.4	37.9	39.8	44.2	46.7	47.0	2.68	10.05	0.001
Energy drinks intake ^†^ (more than 3 times/week)	Overall	-	-	-	-	-	3.3	3.3	3.9	8.0	-	12.2	1.93	5.16	0.014
	Boys	-	-	-	-	-	4.3	3.9	4.3	8.9	-	12.8	1.89	4.32	0.023
	Girls	-	-	-	-	-	2.2	2.6	3.5	7.0	-	11.6	1.99	6.46	0.008

Note: Sweet drink intake and energy drink intake have been surveyed since 2014. The prevalence was calculated by applying sampling weight. Annual prevalence from 2009 to 2019 was used for trend analysis. *p*-value refers to the statistical significance of test results for the annual change in prevalence. ^†^ In the case of not having prevalence in 2018 due to survey questionnaire change, the estimated annual change was calculated with the available prevalence rate.

**Table 4 ijerph-19-05672-t004:** Trends of Physical Activity among Adolescents from 2009 to 2019 by Sex.

Variables	Category	Year											b	t	*p*
		2009	2010	2011	2012	2013	2014	2015	2016	2017	2018	2019			
Vigorous-intensity physical activity	Overall	31.6	33.0	34.1	33.6	35.9	37.2	37.9	37.7	37.3	37.8	32.0	0.43	3.59	0.003
	Boys	43.3	45.3	46.9	46.3	47.3	48.7	50.8	49.4	49.6	50.7	44.8	0.42	3.11	0.008
	Girls	18.4	19.3	20.0	19.5	23.4	24.6	23.8	24.9	24.0	23.7	18.0	0.50	3.72	0.003
60 min of physical activity a day (more than 5 days/week)	Overall	10.9	10.0	10.8	12.0	12.6	13.8	14.2	13.1	13.8	13.9	14.7	0.43	6.40	<0.001
	Boys	15.7	14.5	15.8	17.3	17.8	19.2	20.5	18.8	19.5	20.3	21.5	0.63	7.47	<0.001
	Girls	5.4	4.9	5.2	6.1	6.9	8.0	7.4	7.0	7.5	7.1	7.3	0.25	3.90	0.004
Muscle-strengthening exercise (more than 3 days/week)	Overall	20.1	20.9	20.0	19.2	20.1	22.1	22.1	20.8	22.8	23.4	21.9	0.15	2.34	0.036
	Boys	29.1	30.5	29.5	28.2	29.6	32.0	32.9	30.9	33.7	35.2	33.4	0.30	3.21	0.007
	Girls	9.9	10.0	9.3	9.2	9.8	11.3	10.3	9.8	11.0	10.5	9.4	0.03	0.59	0.568
Sedentary time per day (minutes)	Overall	-	-	-	-	1057.2	1071.4	1086.1	1099.9	1102.0	1154.3	1160.2	17.53	7.94	0.001
	Boys	-	-	-	-	973.8	985.0	998.2	1009.7	1023.7	1073.0	1099.0	20.61	7.57	0.001
	Girls	-	-	-	-	1148.5	1165.2	1180.9	1197.5	1186.5	1242.9	1225.8	14.03	5.22	0.003

Note: The prevalence was calculated by applying sampling weight. Annual prevalence from 2009 to 2019 were used for trend analysis. *p*-value refers to the statistical significance of test results for the annual change in prevalence. Sedentary time per day has been surveyed since 2013.

## Data Availability

The data that support the findings of this study are available from the corresponding author upon reasonable request.

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
