# Peer review of "Trends in Obesity and Obesity-Related Risk Factors among Adolescents in Korea from 2009 to 2019"

_ijerph, 2022, doi:10.3390/ijerph19095672_

Round 1
Reviewer 1 Report
Obesity in adolescents is a major public health problem and has been increasing in recent years or decades. The following comments are sent to the authors for their consideration.
- The study used public data between 2009 and 2019. The response rates over the years were decreasing, however, the authors note that the response rate for 2019 was 95.3%, however, this rate is due to obtained from the data of 2009 and not of the previous year.
- The surveys were self-administered. What effect could this have on the results?
- Why use the tables of the Ministry of Health and Welfare and the Korean Academy of Pediatrics and not the tables and graphs of the World Health Organization in order to have comparison data within and between countries? How were the adolescents asked for information about their anthropometric data? Were they asked to weigh and measure themselves? Since according to studies have shown that people with obesity may not underreport information, in this case anthropometric data, food and physical activity.
- What were the cut-off points for the diagnosis of obesity?
- Was information obtained on milk with added sugars?
- Have the surveys of eating habits and physical activity been validated?
- Was the data obtained from adolescents at the national level? Or just some states?
- Was information obtained such as socioeconomic level that could be a factor for the development of obesity?
- Statistical analysis. What statistical tests were used to evaluate the changes in the prevalence of obesity in the different years? It is not mentioned if information referring to incomplete data, outliers, was excluded. How many were excluded at what moments in time? What do you mean by weighted weight?
- Why was the prevalence of overweight not obtained? They are children who are also at risk of developing obesity in the short term.
- Table 1. Does the p-value refer to a trend test? Place the information at the bottom of the table and in the statistical analysis section.
- Table 2. Does the p-value refer to a trend test? Place the information at the bottom of the table and in the statistical analysis section. Was there a difference between boys and girls in the information obtained?
- Table 2 and 3. Does the p-value refer to a trend test? Place the information at the bottom of the table and in the statistical analysis section. Was there a difference between boys and girls in the information obtained?
- The objective of the study was to evaluate the risk factors related to obesity in adolescents. In the statistical analysis, no association was made between the factors of diet, physical activity and sedentary lifestyle with the development of obesity. It is suggested to make association models.
- The article notes that the prevalence of obesity was higher in boys than in girls for all years, however, the authors do not show statistical evidence of such differences.
- The authors state in their conclusion: “This study found different factors influencing the development of suicidal ideation and transition from suicidal ideation to attempt.” However, this conclusion has nothing to do with this study.
Author Response
Dear Reviewer
We wish to thank you for your thoughtful comments and valuable feedback on the manuscript entitled “Trends in obesity and obesity-related risk factors among Adolescent in Korea from 2009 to 2019.” We would like to resubmit the revised manuscript for publication in the International Journal of Environmental Research and Public Health.
We have tried to revise the manuscript according to your suggestions and rewrote or rephrased sections to improve clarity. For your convenience, we have used red font for the revisions. Please find the following revisions according to reviewer’s comments.
Further, I believe that this revised paper will be of interest to the readership of the International Journal of Environmental Research and Public Health. Thank you for your consideration. I look forward to hearing from you.

Reviewer 2 Report
Dear Authors,
Thank you for your manuscript. The paper is interesting, well-written, the topic is important for public health. I have only few comments.
Please provide study aim at the end of the Introduction.
In the section 2.1, please provide a more detailed information on sampling methods and explanation how the country-representative sample was selected.
Please provide a reference for the cut-off of 3 days/week and more of vigorous PA.
Please provide study limitations.
Author Response

(The authors gave the same response as above.)

Reviewer 3 Report
This is an interesting article as it describes trends in obesity and its risk factors. However, it is only an ecological descriptive study. I wonder if they had access to the raw data which could enable them to analyze the association between the change in BMI and the nutrition behavior and physical activity or sedentary behaviors. This kind of analysis could have contributed to a better understanding and better direct to intervention.
It is a pity that Hight and weight were self-reported and not measured. However, I would expect to see some discussion related to the possible bias of self-reported hight and weight, especially in relation to the differences between boys' and girls' report. The author tried to explain that difference, but they don't refer to the source of these data. There is literature on these differences in reports. The authors suggested that " we need to consider the sex differences in obesity and the related factors of obesity prevention and management…" but they should discuss the possibility that it is not the real difference, but it could only be a difference in perceived hight and weight.
In the methods section, the statistical analysis should include much more details. The authors write that "The data analysis results of this study show the effect of correcting the weight variables". This is not clear. Did they control for some variables? Which variables were considered?
The same problem continues to the tables where they refer to weighted %. They should explain both in the data analysis and as a footnote in the table what does it means.
The conclusion section starts with "This study found different factors influencing the development of suicidal ideation and 260 transition from suicidal ideation to attempt". I assume that it is a typing mistake. It probably refers to another study.
Author Response

(The authors gave the same response as above.)

Round 2
Reviewer 1 Report
The authors have responded and made the necessary modifications to the comments sent for their consideration.
Author Response

(The authors gave the same response as above.)

Reviewer 3 Report
In relation to my previous comment on the possible bias of perceived wright & height between boys and girls, the authors refer to it in the limitations of the study. In my opinion, this should be discussed as one of the possible explanations to the differences between boys and girls and not only in the limitations.
In the discussion the authors mentioned studies which demonstrate the association between diet and adolescents' obesity. I wonder why they could not analyze this in their study. They should explain that in the article. However, as the title of the article states as well as the written study objectives – this is a descriptive study which presents only trends of obesity and trends of risk factors. The association between the two is not one of study objectives. To my opinion, they should explain why.
Author Response

(The authors gave the same response as above.)
